# AD-NLP: A Benchmark for Anomaly Detection in Natural Language Processing

**Matei Bejan**[*][†]
University of Bucharest
matei.bejan@s.unibuc.ro

**Andrei Manolache**[*]
Bitdefender
University of Stuttgart
amanolache@bitdefender.com

**Marius Popescu**
University of Bucharest
popescunmarius@gmail.com

## Abstract

Deep learning models have reignited the interest in Anomaly Detection research in recent years. Methods for Anomaly Detection in text have shown strong empirical results on ad-hoc anomaly setups that are usually made by downsampling some classes of a labeled dataset. This can lead to reproducibility issues and models that are biased toward detecting particular anomalies while failing to recognize them in more sophisticated scenarios. In the present work, we provide a unified benchmark for detecting various types of anomalies, focusing on problems that can be naturally formulated as Anomaly Detection in text, ranging from syntax to stylistics. In this way, we are hoping to facilitate research in Text Anomaly Detection. We also evaluate and analyze two strong shallow baselines, as well as two of the current state-of-the-art neural approaches, providing insights into the knowledge the neural models are learning when performing the anomaly detection task. We provide the code for evaluation, downloading, and preprocessing the dataset at https://github.com/mateibejan1/ad-nlp/.

## 1 Introduction

An anomaly, sometimes referred to as an outlier, discordant, or novelty, can be intuitively described as an observation that appears to be inconsistent with the remainder of that set of data (Ord, 1996) to the degree that it arouses suspicion (Hawkins, 1980). Such an observation can be described as being atypical, irregular, erroneous, or simply strange (Ruff et al., 2020). Anomalies are relative to the data distribution at hand and can be perceived as having characteristics that are not definitory to the vast majority of the population. Researchers have been interested in the Anomaly Detection (AD) problem for several decades (Knorr and Ng, 1997; Chandola et al., 2009; Aggarwal and Reddy, 2014), some of the first formal mentions of "discordant observations" going back to the 19th century (Edgeworth, 1887). Both general anomaly detectors (Schölkopf et al., 1999; Liu et al., 2008; Manevitz and Yousef, 2002; Ruff et al., 2018) and narrow-scoped ones (Wang et al., 2019c,b; Ruff et al., 2019; Manolache et al., 2021) have shown promising results in various domains, such as network monitoring (Stolfo et al., 2000; Sharafaldin et al., 2018; Radford et al., 2018), time series (Braei and Wagner, 2020), computer vision (Wang et al., 2019c; Yu et al., 2020), and natural language processing (Ruff et al., 2019; Manolache et al., 2021).

The advent of deep learning methods for detecting anomalies amplified the interest in the field, producing a great variety of models such as ones based on generative networks (Wang et al., 2019b; Zhou and Paffenroth, 2017), self-supervised learning (Wang et al., 2019c; Manolache et al., 2021), or one-class classification (Ruff et al., 2018). Although the field has seen a sprout of activity, most of the introduced datasets for Anomaly Detection are being specifically crafted for Computer Vision, Intrusion Detection Systems (Sharafaldin et al., 2018) or network traffic (Stolfo et al., 2000; Radford et al., 2018). Approaches in NLP are usually benchmarked on ad-hoc setups, typically by making use of an annotated dataset and downsampling some classes to produce outliers (Wang et al., 2019c; Ruff et al., 2019; Manolache et al., 2021). This does not represent an impediment in computer vision, due to the large number and variety of annotated datasets. In natural language processing, however, deciding what an anomaly is and what it is not is a delicate matter. This problem is rooted in the intrinsic complexity of the text: the

---

[*]These authors contributed equally.
[†]Corresponding author.

form the language of a text takes is influenced not only by the style and lexicon of the author but also by the literary or historical era it was written in and its purpose (e.g., newspaper article, novel, letter, satire, etc.). Thus, anomalies can be defined through a multitude of factors, ranging from the concrete syntax, style, and grammar, to the more abstract semantics, metaphorical meaning, and contextual topic. A notable aspect of certain linguistic phenomena that can be effectively analyzed with Anomaly Detection (AD) is the challenge posed by the lack of clear negative examples. For instance, while there are countless texts that were not written by Shakespeare, determining which ones serve as a good representation of non-Shakespearean texts can be tricky. In a similar vein, identifying a typical non-scientific text can also be challenging.

We introduce **AD-NLP**: an anomaly detection setup for Natural Language Processing which can be used to benchmark AD systems on *syntactic anomalies*, *semantic anomalies*, *pragmatic anomalies* and *stylistic anomalies*, by aggregating different tasks and treating them as a general anomaly detection task, such as: *sentence acceptability detection*, *topic detection*, *metaphor detection* and *autorship detection*. Furthermore, we experiment with two strong classical baselines and two recent deep methods for detecting anomalies in text and make observations on both the quantitative and qualitative results of the models.

In the following, we will use the terms "anomaly" and "outlier" interchangeably to refer to the same concept of divergent observation with respect to the overall data distribution.

The paper is organized as follows: Section 2 covers related work. In Section 3, we outline our task definition, data collection approach, and datasets. Sections 4 and 5 delve into the models we used and our experimental assessment. Finally, we wrap up with our conclusions in Section 6.

## 2 Related Work

### 2.1 AD Datasets

Most of the AD benchmarks were historically used in the domain of Anomaly-based Network Intrusion Detection Systems (A-NIDS). Some of these datasets are synthetic, such as KDD99 (Stolfo et al., 2000), CIC-IDS (Sharafaldin et al., 2018) and the LANL log dataset (Turcotte et al., 2018), while others are obtained using honeypots, like the Kyoto IDS dataset (Song et al., 2011).

In recent years, there has been a focus on Computer Vision, especially on video anomaly detection that uses such benchmarks as Avenue (Lu et al., 2013), ShanghaiTech (Zhang et al., 2016) and UCSD Ped 2 (Mahadevan et al., 2010).

In NLP we note TAP-DLND 2.0 (Tirthankar Ghosal, 2022), a document-level novelty classification dataset, which focuses on semantic and syntactic novelty and tests multiple baselines on a singular task. All datasets of the TAP benchmark are in the newswire domain, we wish to offer a larger variety in terms of anomaly types. On top of this, TAP-DLND 2.0 focuses on detecting the degree of novelty of paraphrased or plagiarized text with respect to one or multiple ground truth texts, while we target out-of-distribution samples compared to an overarching distribution.

Additionally, out-of-distribution detection setups (Arora et al., 2021) can be used to construct ad-hoc outliers in the validation or test data, albeit these distribution-shifted samples are artificial.

### 2.2 NLP Datasets

Anomaly detection can be viewed as a particular case of the one-class classification setting. Classification datasets can be ideal for AD tasks due to their ease of being adapted for one-class unsupervised classification (Ruff et al., 2019; Manolache et al., 2021). Various NLP classification benchmarks are widely used, their purpose varying and including news topic detection tasks (Sam Dobbins, 1987; Lang, Ken and Rennie, Jason, 2008; Zhang et al., 2015), sentiment analysis (Maas et al., 2011; Socher et al., 2013; He and McAuley, 2016) or authorship verification (Bevendorff et al., 2020).

Recently, more general NLP datasets that contain multi-task challenges have emerged due to the desire to obtain general NLP models. **decaNLP** (McCann et al., 2018) provides a general framework for multitask learning as question answering and is proposing ten tasks together with a leaderboard to facilitate research in areas such as multitask learning, transfer learning, and general purpose question answering. **GLUE** (Wang et al., 2018) is a multi-task benchmark and analysis platform for Natural Language Understanding. GLUE provides a suite of nine sentences or sentence-pair NLU tasks, an evaluation leaderboard, and a so-called "diagnostic evaluation dataset". The performance on the GLUE benchmark has surpassed the

Table 1: Data statistics for our benchmark. By the "Several" domain we denote a blend of Politics, Science, Fiction, Academia, and News. The Avg #Words column denotes each dataset's average number of words per sample.

| Corpus | Train | Test | Avg #Words | Domain | Anomaly |
|---|---|---|---|---|---|
| 20Newsgroups | 10.996 | 8.819 | 92 | News | Semantic |
| AGNews | 120.000 | 7.600 | 31 | News | Semantic |
| COLA | 8.551 | 1.043 | 8 | Linguistics | Syntactic |
| VUA | 8.485 | 3.637 | 15 | Several | Pragmatic |
| Song Genres | 15.120 | 3.780 | 249 | Music | Stylistic |
| Gutenberg Categories | 5.000 | 1.000 | 618 | Several | Semantic |
| Gutenberg Authors | 4.765 | 1.000 | 619 | Fiction+Politics | Stylistic |

level of non-human experts in just twelve months, thus leading to the release of an updated variant called **SuperGLUE** (Wang et al., 2019a).

## 3 Data

### 3.1 Task Definition

**Problem Setting** Our benchmark is aimed at addressing a broad spectrum of anomaly detection scenarios. In this regard, we concentrate on three crucial elements for our benchmark: diversity in the domain of natural language where anomalies occur, diversity within each of these domains (see Table 2), covering various settings that might arise within the same NLP domain, and diversity in terms of sample counts. The final aspect explores the variation in both train and test sample counts across datasets, as well as the variation in sample numbers for each class within each dataset.

**Dataset Properties** Firstly, our benchmark covers four outlier classes, as can be seen in Table 2. On top of this, it also aims at delivering intra-class variety by supplying multiple datasets for our content category. We believe this is important so as to not lock an anomaly class to a specific instance of outlier distribution. As an example, content anomalies can appear in news data through a minority of articles with a diverging topic, whereas when it comes to music genres, lyricism, or fiction writing, the outliers can present a cluster of multiple similar subjects which are different from the ones of the majority class. Some dataset statistics are available in Table 1, with more detailed information in our benchmark's datasheet [1]. Similarly, training

---

[1] https://github.com/mateibejan1/ad-nlp/blob/main/datasheet.md

and validation code can be obtained through our GitHub repository.

### 3.2 Data Collection

Our data acquisition methodology was designed with the clear goal of providing a large diversity of what we can define as anomalies. This has been done so the data mimics the in-the-wild distribution of classes as well as possible. Our benchmark consists of four already-available datasets: 20Newsgroups (Lang, Ken and Rennie, Jason, 2008), AG News (Zhang et al., 2015), CoLA (Warstadt et al., 2018) and VUA (Steen et al., 2010), as well as novel datasets: Song Genres, Gutenberg Categories, and Gutenberg Authors. Table 3 provides an overview of the data through examples of outliers and inliers.

### 3.3 Available Datasets

We selected a set of representative existing datasets due to their prior utilization in literature (Ruff et al., 2019; Manolache et al., 2021). The 20Newsgroups and AGNews datasets have been frequently used in various experimental setups, hence, we included them to ensure comprehensiveness. Additionally, we incorporated COLA for syntactic anomaly detection and VUA for metaphor detection.

**20Newsgroups.** The 20Newsgroups dataset (Lang, Ken and Rennie, Jason, 2008) amounts to almost 20,000 news documents clustered into twenty groups, each of which corresponds to a different topic. Some newsgroups are closely related, such as *comp.sys.ibm.pc.hardware* and *comp.sys.mac.hardware*, while others are highly unconnected, namely, *misc.forsale* and *soc.religion.christian*.

Table 2: The outlier classes contained in our benchmark and a succinct definition of their data type.

| Domain | Description |
| --- | --- |
| **Syntactic** | A pronounced divergence in the arrangement of words and phrases. |
| **Semantic** | A difference in the subject or content appearing throughout the sample. |
| **Pragmatic** | The presence of metaphors or figures of speech in the sample. |
| **Stylistic** | A distinctive manner of expression, including but not limited to: repetition of verbs or phrases, use of stopwords or punctuation. |

We have extracted six classes from the initial twenty: computer, recreation, science, miscellaneous, politics, and religion, as done in (Ruff et al., 2019). Each category is represented by a range of 577-2.856 training samples and 382-1.909 validation samples. Despite the relatively small size of the dataset, its classical relevance for NLP tasks prompted us to include it in our analysis.

**AG News.** The AG News dataset (Zhang et al., 2015) encompasses 496.835 categorized news articles from over 2.000 news sources. This topic classification corpus was gathered from multiple news sources for over a year. Out of all the classes, we have chosen the four largest classes in the dataset: business, sci, sports, and world. We are using the full 30.000 training samples per class, selecting only the title and description fields.

**CoLA.** CoLA (Warstadt et al., 2018) consists of a corpus of 10.657 sentences from 23 linguistics publications, books, and articles, annotated as being grammatically acceptable (in other words, grammatically correct) or not. Each sample is a sequence of words annotated with whether it is a grammatical English sentence. The public version of this dataset we used contains 8.551 sentences belonging to the training set and 1.043 sentences as part of the development set. As the test is not publicly available, we have used the development set as a de-facto test set for our work.

**VUA.** VUA (Steen et al., 2010) consists of 117 fragments sampled across four genres from the British National Corpus: Academic, News, Conversation, and Fiction and contain word-level all content-word metaphors annotations. The train set contains 12.122 lines or sentences and is the only publicly available subset of VUA. Under these circumstances, we applied an 80-20 train-test split on this solely open data subset and produced 8.485 train samples and 3.637 test samples.

The data is annotated according to the MIPVU procedure described by its authors. As a conse-

quence, the words annotated as metaphors have been prefixed with the "M_" string in the original annotation setting. To transform this initial problem setup of text segmentation into one of anomaly detection and for the data to comply with our methodology, we removed the word-level annotations and instead labeled the whole sentences as containing a metaphor or not.

### 3.4 Newly-Proposed Datasets

We introduce three new datasets: Song Genres, Gutenberg Categories, and Gutenberg Authors. The latter two were extracted from the Project Gutenberg website [2]. We scraped the entire website and parsed all bookshelves, which stored the book texts, their authors, titles, and the category in which Project Gutenberg placed them. We annotated the books for said category. The result is a corpus of over 15.000 literary texts[3], along with their authors, titles, and titles and bookshelves (a term that Gutenberg maintainers use for categories). We then filtered this dataset to produce **Gutenberg Categories** and **Gutenberg Authors**.

**Song Genres.** The Song Lyrics is a dataset[4] composed of four sources and consists of over 290.000 multilingual song lyrics and their respective genres. The initial data was forwarded from the 2018 Textract Hackathon[5]. This was enhanced with data collected from three other datasets from Kaggle: *150K Lyrics Labeled with Spotify Valence*, *dataset lyrics musics*, and *AZLyrics song lyrics*.

To deal with the lack of labels, we have built a labeling system using the *spotipy* library, which uses the Spotify API to retrieve an Artist's genre. The Spotify API returns a list of genres for one artist, so we consider the mode of that list to be the

---

[2]https://www.gutenberg.org/
[3]https://www.kaggle.com/mateibejan/15000-gutenberg-books
[4]https://www.kaggle.com/mateibejan/multilingual-lyrics-for-genre-classification
[5]https://www.sparktech.ro/textract-2018/

Table 3: Data samples for each anomaly domain included in our benchmark. Apart from *Pragmatic* and *Syntactic*, we used only a chunk of the sample. We did not mention the inlier class in this table, as the majority class is constructed by clustering all non-outlier classes from the dataset as inliers.

| Domain | Outlier Class | Outlier Sample | Inlier Sample |
|---|---|---|---|
| **Semantic** | computers | Apple has patented their implementation of regions, which presumably includes the internal data structure. | I am looking for an inexpensive motorcycle, nothing fancy, have to be able to do all maintenance my self. |
| **Syntactic** | Gramatically Unacceptable | They caused him to become angry by making him. | Bill coughed his way out of the restaurant. |
| **Pragmatic** | Metaphor | Mr Franklin went there at the end of the 1970s, after the collapse of Keyser Ullman. | It would be a criticism if I was doing it to impoverish myself. |
| **Stylistic** | Arthur Conan Doyle | You may place considerable confidence in Mr. Holmes, sir, said the police agent loftily. | Mars, I scarcely need remind the reader, revolves around the sun at a mean distance of 140,000,000 miles. |

dominant genre of the lyrics of said artist. Additionally, we used the *langdetect* library to label the lyrics with a language automatically. In total, the lyrics come in 34 languages. Please note that we have only used the lyrics as our training data, with their respective genres as labels, leaving aside the corresponding artist, year, or song name. We've applied this procedure for all the data apart from the original 2018 Textract data.

From the original dataset, we have curated our subsequent **Song Genres** subset, which encompasses nine genres: Pop, Hip-Hop, Electronic, Indie, Rock, Metal, Country, Folk, and Jazz. Song Genres is designed to present an anomaly setup where crucial data aspects (e.g., melody, rhythm, etc.) are obscured or absent. This accentuates the necessity of discerning subtle text variations in songs to distinguish between different groups. Through this, we aim to foster the advancement of more robust models for AD in NLP.

**Gutenberg Categories.** The initial subset derived from the original Gutenberg data is termed as the Gutenberg Categories dataset. It comprises texts corresponding to 10 categories from the Gutenberg project website: Biology, Botany, United States Central Intelligence Agency, Canada, Detective, Historical, Mystery, Science, Children's, and Harvard. It's important to note that the categories are not inherently distinct by nature. Some, like CIA and Children's, are expected to be eas-

ily distinguishable, while others, such as Biology and Botany or Science and Harvard might exhibit significant overlap. We specifically included the CIA category to offer a class that stands distinctly separable from the rest in the text distribution.

We have selected 500 train samples from each class for the train set, and 100 samples per class for the test set. The samples have been extracted from multiple authors for each category, as to offer a wider distribution of styles, syntax, and grammar.

**Gutenberg Authors.** The Gutenberg Authors dataset represents our second subset generated from the project Gutenberg data. It comprises the texts respective to 10 authors: Arthur Conan Doyle, Charles Dickens, Charles Darwin, Mark Twain, Edgar Allan Poe, Walter Scott, United States Central Intelligence Agency, H. G. Wells, L. M. Montgomery, and Agatha Christie. Again, we aimed at providing different levels of complexity throughout our data, the reason for which we included authors whose novels are within the same genre, namely Doyle and Christie, as well as those writing about the same historical era or similar historical events, such as Twain and Dickens, and female authors which supposedly share a common sensibility, meaning Montgomery and Christie. We've added the CIA and Darwin classes with the same purpose as for **Gutenberg Categories**.

We have sampled between 400 and 500 train text chunks for each author and 100 test samples.

The samples have been extracted through multiple books for each author we consider for this experiment. This has been done to avoid the possibility of the event in which a particular anomaly class might be locked into a repetitive word or phrasing, e.g., character names (Sherlock Holmes, Huckleberry Finn, etc.), places (London, Washington, etc.) or simply by the sample length.

## 4 Models

### 4.1 Experimental Methodology

Our methodology consists of creating multiple data splits for each dataset within our benchmark, running our models on all splits, and finally aggregating the results. By data split, we refer to a separation of classes into one inlier class and a cluster of classes that are considered to be outliers in this setup. To achieve this, we iterate through all the classes for every dataset and, at each iteration, choose one of them as the inlier, while the rest are treated as outliers by the models. Through this, we achieve an important objective: we unfold an exhaustive series of experiments over all possible combinations of outliers and inliers, outlining which of the former are the most prominent and which are the hardest to detect. We ran a hyperparameter search for the two classical models on each split, thus finding the best parameters for detecting each outlier choice. The deep models were trained with a limited set of hyperparameters, as can be seen in Subsections 4.2 and 4.3.

**Evaluation Metrics.** We use AUROC in Table 4, as well as AUPR-In, and AUPR-Out in Table 6 and Supplementary Tables 7 and 8. AUROC (Area Under the Receiver Operating Characteristic) is the area under the curve where the false positive rate is on the X-axis and the true positive rate on the Y-axis. AUPR-In (Area Under the Precision-Recall for Inliers) is the area under the curve where the recall is on the X-axis and the precision is on the Y-axis. AUPR-Out (Area Under the Precision-Recall for Outliers) has the same definition as AUPR-In but is computed on inverted labels.

### 4.2 Classical approaches

The SVM classifier is a versatile model adaptable for outlier detection tasks, known as the **One Class Support Vector Machine**, as detailed in (Schölkopf et al., 1999). The OC-SVM aims to learn from an inlier class, designating test samples as anomalies when they deviate from the train-

ing dataset. The **Isolation Forest** technique is another outlier detection approach, drawing inspiration from the Random Forest model, as described in (Liu et al., 2008). In an n-dimensional space, inliers typically form denser clusters, whereas outliers tend to be more dispersed.

To optimize the performance of traditional methods, we conducted a comprehensive hyperparameter tuning. For the OC-SVM, we explored different kernels, namely rbf, polynomial, and linear, and assessed a range of $\nu$ values: $\nu \in 0.05, 0.1, 0.2, 0.5$. For the Isolation Forest, we evaluated various numbers of estimators, specifically $64, 100, 128, 256$. For both models, we compared the effectiveness of two embedding methods: FastText and GloVe, each with an embedding size of 300.

### 4.3 Neural approaches

**CVDD.** Context Vector Data Description (CVDD) (Ruff et al., 2019) is a method that takes advantage of pre-trained word embeddings to perform AD on text. CVDD jointly learns the so-called "context vectors" and a multi-head self-attention mechanism that projects the word representations near these "context vectors" by minimizing the cosine distance between them.

CVDD allows the disentanglement of the context vectors such that they can provide more interpretable results and penalize non-orthogonal context vectors. The resulting projection function and context vectors act like a clustering method and cluster centroids, respectively. Anomalies are detected based on the mean sample distance from the sequence projection to the context vectors. We only search for the optimal number of context vectors like in (Ruff et al., 2019) ($c \in \{3, 5, 10\}$) and report the best performing models.

**DATE.** Detecting Anomalies in Text using ELECTRA (DATE) is an approach that uses self-supervision for training Transformer networks using two pretext tasks tailored for detecting anomalies - Replaced Token Detection (RTD) (Clark et al., 2020) and Replaced Mask Detection (RMD). The method uses a generator to sample masked tokens and a discriminator to identify the replaced tokens and the masking patterns. The generator can be any distribution over the vocabulary. The discriminator is a BERT (Devlin et al., 2018) model trained from scratch. The Replaced Token Detection head ouputs a token-wise anomaly score which

Table 4: Best AUROC scores for each model and each split. We also provide the mean and standard deviations between the splits for all datasets apart from COLA and VUA, where we use a single split as our inlier class.

| | Inlier Class | iForest | OCSVM | CVDD | **DATE** |
|---|---|---|---|---|---|
| **20 News** | comp | 75.6 | 76.8 | 51.9 | **92.1** |
| | rec | 68.0 | 74.1 | 55.8 | **83.4** |
| | sci | 66.7 | **73.5** | 49.2 | 69.7 |
| | misc | 60.8 | 64.0 | 59.6 | **86.0** |
| | pol | 53.0 | 60.9 | 63.2 | **81.9** |
| | rel | 74.8 | 78.7 | 53.8 | **86.1** |
| | mean ± std | 66.5 ± 7.8 | 71.3 ± 6.6 | 55.6 ± 4.7 | **83.2 ± 6.8** |
| **AG News** | business | 72.9 | 83.9 | 64.0 | **90.0** |
| | sci | 75.6 | 80.6 | 70.7 | **84.0** |
| | sports | 78.3 | 89.1 | 79.2 | **95.9** |
| | world | 83.8 | 86.0 | 65.8 | **90.1** |
| | mean ± std | 77.6 ± 4.0 | 84.9 ± 3.1 | 69.9 ± 5.9 | **90.0 ± 4.2** |
| **Song Genres** | Indie | 51.5 | 45.6 | 47.1 | **53.7** |
| | Pop | 43.1 | 43.6 | 47.2 | **57.6** |
| | Metal | 55.1 | 43.4 | 46.9 | **51.2** |
| | Hip-Hop | 63.3 | **68.0** | 66.8 | 54.0 |
| | Electronic | 47.8 | 47.2 | 44.6 | **54.0** |
| | Country | 57.8 | 57.7 | 54.6 | **66.7** |
| | Folk | 48.9 | 50.2 | 48.9 | **52.0** |
| | R&B | 57.6 | 56.6 | 55.5 | **70.1** |
| | Rock | 44.7 | 45.8 | 43.5 | **54.8** |
| | Jazz | 53.3 | 53.0 | 45.7 | **70.2** |
| | mean ± std | 52.3 ± 7.8 | 51.1 ± 7.4 | 50.1 ± 6.7 | **58.0 ± 7.2** |
| **Gutenberg Categories** | Detective | 70.9 | 72.2 | 67.5 | **82.5** |
| | Botany | 45.9 | 42.8 | 46.9 | **74.2** |
| | CIA | 88.1 | 94.8 | 85.6 | **99.9** |
| | Mystery | 73.3 | 69.3 | 55.4 | **85.4** |
| | Biology | **83.5** | 82.3 | 79.5 | 81.9 |
| | Children's | 56.9 | 58.8 | 65.0 | **79.7** |
| | Harvard | 70.3 | 79.3 | 66.6 | **82.1** |
| | Canada | 45.5 | 39.7 | 49.0 | **53.3** |
| | Science | 70.4 | 67.2 | 55.4 | **68.0** |
| | Historical | **77.3** | 73.3 | 75.5 | 66.7 |
| | mean ± std | 68.2 ± 13.8 | 68.0 ± 16.2 | 64.6 ± 12.3 | **77.4 ± 12.0** |
| **Gutenberg Authors** | C. Dickens | 83.5 | 85.1 | **90.0** | 35.1 |
| | A.C. Doyle | 53.6 | 41.8 | **55.4** | 34.7 |
| | M. Twain | 85.8 | 82.8 | 78.8 | **93.7** |
| | C. Darwin | 74.0 | 75.4 | 82.9 | **99.8** |
| | W. Scott | 87.1 | 82.1 | **91.0** | 11.0 |
| | A. Christie | **93.4** | 91.2 | 92.5 | 55.2 |
| | E.A. Poe | 71.9 | 72.7 | **78.6** | 48.7 |
| | CIA | 87.9 | 94.7 | 83.0 | **100** |
| | L.M. Montgomery | 78.3 | 84.9 | **94.1** | 61.5 |
| | H.G. Wells | 53.7 | **59.8** | 37.9 | 11.8 |
| | mean ± std | 76.9 ± 13.2 | 77.0 ± 15.0 | **78.4 ± 17.2** | 55.2 ± 32.0 |
| **COLA** | 1 (Acceptable) | 49.9 | 49.2 | 53.3 | **57.2** |
| **VUA** | 0 (Non-Metaphor) | 75.3 | 76.2 | **76.6** | 51.1 |

Table 5: CVDD's ability to cluster semantically related words from the Hip-Hop Music Genre subset. The model effectively identifies variations in verb and pronoun usage, recognizes foreign language terms, and associates specific contexts with obscene terms. We highlight second context as it is the most meaningful one when detecting anomalies.

| Context 1 | **Context 2** | Context 3 | Context 4 | Context 5 |
|-----------|---------------|-----------|-----------|-----------|
| 'm | **livin** | to | the | que |
| i | **turnin** | can | of | mas |
| 're | **shakin** | could | in | como |
| y' | **waitin** | make | 's | porque |
| somebody | **walkin** | will | that | sabe |
| myself | **keepin** | would | is | desde |
| everybody | **slippin** | pray | this | boca |
| obscenity1 | **lickin** | try | surrounds | se |
| obscenity2 | **standin** | come | descends | encontrar |

Table 6: The Area under the Precision-Recall curve for all of the datasets, averaged over the subsets. We highlight the best scores in **bold**. Abbreviations: 20NG - 20Newsgroups, AGN - AGNews, SG - Song Genres, GC - Gutenberg Categories, GA - Gutenberg Authors, CA - COLA, VA - VUA.

| | Metric | OCSVM | iForest | CVDD | DATE |
|---|--------|-------|---------|------|------|
| **20NG** | AUPR-In | **90.8** ± 5.0 | 89.6 ± 5.6 | 85.0 ± 7.3 | 41.4 ± 14.9 |
| | AUPR-Out | 40.1 ± 17.5 | 32.6 ± 13.5 | 22.5 ± 7.7 | **96.0** ± 1.7 |
| **AGN** | AUPR-In | **93.1** ± 1.0 | 88.9 ± 3.3 | 85.5 ± 3.6 | 76.9 ± 9.7 |
| | AUPR-Out | 70.4 ± 6.7 | 57.5 ± 6.9 | 46.1 ± 8.9 | **95.9** ± 1.7 |
| **SG** | AUPR-In | 90.6 ± 2.3 | **90.7** ± 1.9 | 90.3 ± 2.3 | 12.8 ± 3.6 |
| | AUPR-Out | 12.3 ± 7.0 | 11.7 ± 4.0 | 10.2 ± 2.3 | **92.3** ± 1.9 |
| **GC** | AUPR-In | 94.0 ± 4.8 | **94.1** ± 4.3 | 93.9 ± 3.2 | 36.8 ± 26.1 |
| | AUPR-Out | 25.1 ± 24.0 | 22.6 ± 16.3 | 19.4 ± 11.2 | **96.3** ± 2.7 |
| **GA** | AUPR-In | 96.3 ± 2.5 | **96.4** ± 2.2 | 96.3 ± 4.1 | 48.5 ± 33.5 |
| | AUPR-Out | 39.4 ± 25.3 | 36.8 ± 21.1 | 37.9 ± 17.7 | **97.2** ± 3.5 |
| **CA** | AUPR-In | 31.8 ± N/A | 31.6 ± N/A | 34.2 ± N/A | **36.4** ± N/A |
| | AUPR-Out | 68.0 ± N/A | 66.9 ± N/A | 69.9 ± N/A | **73.5** ± N/A |
| **VA** | AUPR-In | 77.2 ± N/A | **78.1** ± N/A | 76.1 ± N/A | 55.7 ± N/A |
| | AUPR-Out | 73.3 ± N/A | 74.8 ± N/A | **75.0** ± N/A | 48.4 ± N/A |

the search methodology mentioned in Sec. 4. The models were trained on an *inlier class*, and every other class was considered anomalous at test time.

The results of the experiments are in Tables 4 and 6, calculated as averages over all splits for each dataset. We can observe that DATE has achieved the best AUROC scores on six out of our seven datasets, with a total average score of 71.11%, followed by CVDD, which scored the best on VUA. The average AUROC scores are almost equal across the three models: OC-SVM - 64.9%, IsoForest - 63.2%, CVDD - 64.3%. Interestingly, every other model is close to 50% AUROC on COLA. Results on 20Newsgroups and AGNews are similar across all splits; thus, it makes sense to consider either split as the inlier.

The Area under the Precision-Recall Curve (AUPR) results are summarized in Table 6, with detailed results for each split being available in the Appendix in Tables 7 and 8, where we provide an exhaustive table with the metrics for every split and for each dataset.

can also be used as an explainability mechanism. The anomalies are then detected by averaging the $PL_{RTD}$ score over the entire sequence.

We use the same experimental setup and hyperparameters as in (Manolache et al., 2021) for the 20Newsgroups and AG News datasets, and the AG News setup for the other datasets.

## 5 Experiments

### 5.1 Quantitative results

We test the performance of all models on all the datasets within our benchmark and report the scores using the best hyperparameters following

### 5.2 Qualitative results

We provide a sample of qualitative results in Figure 1, where we show how DATE can detect semantic nuances for the Botany subset of Gutenberg Categories, and in the Supplementary Figure 2, where we show how DATE can detect stylistic changes when correctly profiling an author. The model is suspicious of certain punctuation and word usage. When scoring an inlier entry, DATE seems to think that some stopword usage is anomalous; this could indicate a change of stylistic approach when an

**Botany Detected Inlier**

thirty flowers on the crossed plants were crossed with ##pol ##len from other crossed plants of the same generation , and the ##t ##wen ##ty - six capsule ##s thus produced contained , on an average , 4 . 3 seeds ; whilst thirty flowers on the self - fe ##rti ##lis ##ed plants , fe ##rti ##lis ##ed with the ##pol ##len from the same flower , produced twenty - three capsule ##s , each ##con ##tain ##ing 4 . 3 seeds . thus the average number of seeds in the crossed ##cap ##sul ##es was to that in the self - fe ##rti ##lis ##ed capsule ##s as to . ah ##und ##red of the crossed seeds weighed [SEP]

**Botany Detected Outlier**

it would have been an old look for a child of ##t ##we ##l ##ve , and sara crewe was only seven . the fact was , however , that she ##was always dreaming and thinking odd things and could not herself ##rem ##em ##ber any time when she had not been thinking things about grown - up ##pe ##ople and the world they belonged to . she felt as if she had lived along , long time . t this moment she was remembering the voyage she had just made from ##bo ##mba ##y with her father , captain crewe . she was thinking of the big ##ship , of the las ##cars passing si [SEP]

Figure 1: We provide a qualitative examples for outliers and inliers detected by DATE for the Botany subset of the Guttemberg Categories dataset. A stronger red highlight indicates a greater anomaly score. We can see that, in the case of the inlier sample, there is no strong indication of abnormalities, since the contents relate to Botany. For the outlier sample, words such as "ship" and "cars" are detected as being anomalous.

author is writing texts in registers.

We show in Table 5 how CVDD creates clusters of semantically similar words when trained on the Hip-Hop Music Genre subset. The context vectors act as topic centroids. CVDD cannot only distinguish between colloquial usage of verbs, pronouns, and foreign languages but also associates unreproducible words (e.g., obscene, insulting, etc.) with certain contexts.

## 6 Conclusions

We introduce **AD-NLP**, a benchmark for anomaly detection in text over an extensive assortment of outlier scenarios, covering syntactic, semantic, pragmatic, and stylistic language anomalies. Additionally, we introduce three new datasets as part of **AD-NLP**: Song Genres, Gutenberg Categories, and Gutenberg Authors. Song Genres provides a complex setting in which part of information about the data has been obscured, enforcing a distinct focus on the subtle differences between texts, while the two datasets derived from the Gutenberg data are meant to provide variety on multiple levels: syntax, style, genre, and literary movement. We find that anomalies that depend solely on semantic or stylistic aspects of the text are easier to recognize, whereas those that only partially depend on the text,

like song lyrics, are harder to detect and separate. We have also disclosed our results on various models and found out that the neural models react well to domain-specific words, author idiosyncrasies and punctuation as being anomalous or not. We hope that the proposed benchmark and tools will facilitate research in Text Anomaly Detection.

## 7 Limitations & Further Work

Some of our more simple baselines managed to outperform more sophisticated anomaly detection methods in some scenarios. For instance, we did observe that on datasets such as COLA and VUA, both CVDD and DATE obtain weak results - as an example, in Tables 7 and 8 of the Supplementary Material, we can observe that the OC-SVM and the Isolation Forest outperform DATE on the metaphor detection task from VUA. Moreover, we can observe that for every dataset in the benchmark, there are instances where the Isolation Forest and the OC-SVM outperform the more sophisticated CVDD and DATE methods. Therefore, we believe that it's very important to be able to analyze the limitations and inductive biases on a wide range of scenarios while developing an Anomaly Detection methodology. One way of accomplishing this would be studying AD-NLP at an even more granular level. For example, one could determine various linguistic properties of texts written by Edgar Allan Poe, aiming to discover the reasons behind the poor performance of DATE compared to the other authors, as can be observed in Table 4. Further linguistic analysis would benefit the quality of AD-NLP, and we leave this undertaking for further work.

**Benchmark updates.** We commit to enhancing AD-NLP with future datasets that would expand the intra or inter-domain variety, either by adding new datasets to AD-NLP or scraping, aggregating, and labeling new data off the web ourselves. We also commit to making the dataset more accessible through multiple hosting services, as well as updating our GitHub repository.

## Acknowledgments

AM acknowledges the International Max Planck Research School for Intelligent Systems (IMPRS-IS) for the support provided.

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

# 8  Supplementary Materials

## 8.1  More quantitative results

In Tables 7 and 8 we present the Area Under the Precision-Recall curve (AUPR) results for all our models and datasets. This provides a clear measure of how each model performed across different datasets.

Table 7: Best AUPR-In scores for each model and for each split.

| | Inlier Class | iForest | OCSVM | CVDD | DATE |
|---|---|---|---|---|---|
| **20 News** | comp | 94.5 | 94.6 | 78.8 | **92.1** |
| | rec | 82.9 | **84.6** | 81.6 | 83.4 |
| | sci | 86.5 | 88.7 | **94.7** | 69.7 |
| | misc | 84.1 | 85.8 | **88.9** | 86.0 |
| | pol | 53.0 | **96.1** | 90.4 | 81.9 |
| | rel | 74.8 | **94.6** | 75.7 | 86.1 |
| **AG News** | business | 83.7 | **92.6** | 82.2 | 74.8 |
| | sci | 89.0 | **91.5** | 86.1 | 62.4 |
| | sports | 89.7 | **94.7** | 90.2 | 88.8 |
| | world | 93.2 | **93.7** | 83.1 | 81.9 |
| **Song Genres** | Indie | **89.9** | 89.1 | 88.8 | 10.5 |
| | Pop | 87.4 | 86.4 | **87.7** | 11.6 |
| | Metal | **91.1** | 90.8 | 89.8 | 10.3 |
| | Hip-Hop | 93.3 | 94.1 | **94.8** | 9.3 |
| | Electronic | **89.7** | **89.7** | 88.5 | 11.5 |
| | Country | 92.9 | **93.1** | 92.5 | 15.1 |
| | Folk | 90.1 | **90.4** | 89.9 | 10.0 |
| | R&B | **92.4** | **92.4** | **92.4** | 18.6 |
| | Rock | 88.3 | **88.4** | 87.8 | 11.3 |
| | Jazz | **91.6** | 91.3 | 89.7 | 19.4 |
| **Gutenberg Categories** | Detective | 95.0 | **95.4** | 94.9 | 38.0 |
| | Botany | 87.2 | 86.8 | **89.0** | 59.2 |
| | CIA | 98.0 | 98.8 | 98.0 | **99.9** |
| | Mystery | **96.2** | 95.5 | 92.6 | 32.6 |
| | Biology | **97.6** | 97.5 | 96.8 | 28.1 |
| | Children's | 93.1 | 93.5 | **94.2** | 30.0 |
| | Harvard | 95.5 | **97.0** | 95.4 | 36.5 |
| | Canada | 85.4 | 84.7 | **88.5** | 11.6 |
| | Science | **95.7** | 95.1 | 93.1 | 18.0 |
| | Historical | **96.9** | 96.4 | 96.6 | 13.6 |
| **Gutenberg Authors** | C. Dickens | 97.6 | 98.0 | **98.8** | 35.1 |
| | A.C. Doyle | **93.2** | 90.7 | 92.9 | 7.2 |
| | M. Twain | **98.0** | 97.6 | 96.7 | 61.5 |
| | C. Darwin | 95.3 | 95.7 | 96.4 | **98.8** |
| | W. Scott | 98.3 | 97.6 | **98.9** | 11.0 |
| | A. Christie | 98.9 | 98.5 | **99.1** | 55.2 |
| | E.A. Poe | 94.6 | 94.7 | **96.9** | 48.7 |
| | CIA | 97.5 | 98.3 | 97.4 | **100** |
| | L.M. Montgomery | 97.0 | 97.8 | **99.3** | 55.7 |
| | H.G. Wells | 92.9 | **93.3** | 85.8 | 11.8 |
| **COLA** | 1 (Acceptable) | 31.8 | 31.6 | 34.2 | **36.4** |
| **VUA** | 0 (Non-Metaphor) | 77.2 | **78.1** | 76.1 | 55.7 |

Table 8: Best AUPR-Out scores for each model and for each split.

| | Inlier Class | iForest | OCSVM | CVDD | DATE |
|---|---|---|---|---|---|
| **20 News** | comp | 38.7 | 43.2 | 27.6 | **92.1** |
| | rec | 45.1 | 57.3 | 29.6 | **83.4** |
| | sci | 38.6 | 53.2 | 23.4 | **69.7** |
| | misc | 31.2 | 32.3 | 25.5 | **86.0** |
| | pol | 6.6 | 9.0 | 8.0 | **81.9** |
| | rel | 35.1 | 45.2 | 20.9 | **86.1** |
| **AG News** | business | 49.8 | 67.9 | 38.5 | **96.1** |
| | sci | 52.2 | 61.1 | 46.4 | **93.5** |
| | sports | 61.4 | 79.3 | 58.5 | **98.5** |
| | world | 66.9 | 73.5 | 40.8 | **95.5** |
| **Song Genres** | Indie | 10.6 | 9.1 | 9.1 | **91.6** |
| | Pop | 8.9 | 9.0 | 10.2 | **92.5** |
| | Metal | 12.1 | 11.0 | 9.7 | **89.8** |
| | Hip-Hop | 22.5 | 31.9 | 15.9 | **91.6** |
| | Electronic | 9.3 | 9.1 | 8.5 | **90.9** |
| | Country | 12.4 | 12.1 | 10.9 | **95.0** |
| | Folk | 9.7 | 10.0 | 9.2 | **91.1** |
| | R&B | 12.1 | 11.2 | 11.0 | **94.8** |
| | Rock | 8.8 | 9.5 | 8.2 | **91.0** |
| | Jazz | 10.5 | 10.1 | 8.6 | **95.0** |
| **Gutenberg Categories** | Detective | 22.3 | 23.0 | 17.5 | **97.1** |
| | Botany | 10.8 | 10.8 | 12.2 | **94.2** |
| | CIA | 64.9 | 90.0 | 45.22 | **100.0** |
| | Mystery | 18.2 | 16.0 | 11.7 | **98.2** |
| | Biology | 31.3 | 27.7 | 30.0 | **97.6** |
| | Children's | 11.1 | 12.6 | 16.6 | **97.2** |
| | Harvard | 18.1 | 26.1 | 13.4 | **97.6** |
| | Canada | 9.0 | 8.8 | 10.6 | **90.2** |
| | Science | 17.3 | 15.5 | 10.6 | **95.2** |
| | Historical | 22.1 | 19.0 | 25.4 | **95.5** |
| **Gutenberg Authors** | Charles Dickens | 37.4 | 33.8 | 45.6 | **98.5** |
| | A.C. Doyle | 9.5 | 7.7 | 10.9 | **89.3** |
| | M. Twain | 44.2 | 30.4 | 29.2 | **99.3** |
| | C. Darwin | 29.9 | 33.6 | 45.2 | **99.9** |
| | W. Scott | 39.5 | 28.4 | 48.5 | **94.6** |
| | A. Christie | 74.8 | 69.7 | 48.0 | **99.2** |
| | E.A. Poe | 29.0 | 35.8 | 31.9 | **97.9** |
| | CIA | 65.8 | 91.5 | 47.2 | **100** |
| | L.M. Montgomery | 27.6 | 50.6 | 64.3 | **99.2** |
| | H.G. Wells | 9.8 | 12.4 | 9.2 | **94.1** |
| **COLA** | 1 (Acceptable) | 68.0 | 66.9 | 69.9 | **73.5** |
| **VUA** | 0 (Non-Metaphor) | 73.3 | 74.8 | **75.0** | 48.4 |

# 9 More qualitative examples

In this section, we provide additional qualitative examples to for DATE in Figure 2. These examples are derived from the Gutenberg dataset, focusing specifically on three distinct subsets: Biology, works of Agatha Christie, and CIA-related texts. The aim is to showcase how DATE identifies and highlights anomalous patterns within different authors and styles of text.

**Biology True Positive**

these two ##fa ##cu ##lt ##ies determine all the others . a creature that feels and moves ##re ##qui ##res a stomach to carry food in . food requires instruments tod ##iv ##ide it , liquids to digest it . plants , which do not feel and do not ##mo ##ve , have no need of a stomach , but have roots instead . thus the " animal functions " of feeling and moving determine the character of ##the organs of the second order , the organs of digest ##ion . these int ##hei ##r turn are prior to the organs of circulation , which are a means ##to the end of di ##st ##ri ##bu ##tin [SEP]

**Agata Christie True Negative**

then ##he turned to whitaker ' s alma ##nac ##k to brows ##e upon the statistics of the ##gre ##at european armies . he was rouse ##d from this by the breakfast gong . t breakfast there was no talk of anything but war . hugh was as excited ##as a cat in thunder ##y weather , and the small boys wanted information ##ab ##out flags . the russian and the serbian flag were in dispute , and the ##fl ##ag page of webster ' s dictionary had to be consulted . newspapers and ##lette ##rs were both abnormal ##ly late , and mr . brit ##ling , ti ##rring of supplying ##tri ##vial i [SEP]

**CIA True Negative**

here again we see how much more vigorous the ##cross ##ed plants are than the self - fe ##rti ##lis ##ed . ross ##ed and self - fe ##rti ##lis ##ed plants of the seventh generation . he ##se were raised as here ##to ##for ##e with the following result : - - table 2 / 8 . ip ##omo ##ea pu ##rp ##ure ##a ( seventh generation ) . eight ##s of plants in inches : column : number ( name ) of pot . ol ##um ##n : crossed plants . ol ##um ##n 3 : self - fe

Figure 2: Qualitative examples of DATE on the Gutenberg data. From top to bottom: Biology True Positive sample, Agatha Christie True Negative sample, and CIA True Negative sample. The darker the colour of the highlight, the more greater the anomaly score for the word.