# OpenReview forum: "AD-NLP: A Benchmark for Anomaly Detection in Natural Language Processing"
_EMNLP/2023/Conference — EMNLP 2023 Main_

### Official Review · Reviewer_UgrG · 2023-08-05

**Soundness:** 3

**Excitement:**

3: Ambivalent: It has merits (e.g., it reports state-of-the-art results, the idea is nice), but there are key weaknesses (e.g., it describes incremental work), and it can significantly benefit from another round of revision. However, I won't object to accepting it if my co-reviewers champion it.

**Paper Topic And Main Contributions:**

This paper established a benchmark on anomaly detection in natural language processing, based on 4 existing datasets and 3 newly proposed ones. The authors introduce that the anomaly detection in NLP involve 4 aspects: Syntactic, Semantic, Pragmatic and Stylistic. Also, they evaluate four baselines on these 7 datasets.

Overall, I acknowledge the authors' contributions.
They established four new datasets.
Especially, there is one dataset that may be the first benchmark about Stylistic in NLP.
For these datasets, they made lots of experiments.

**Questions For The Authors:**

Can we summarize the relationship between the characteristics of a method and which type of task it is applicable to, based on existing research results?

**Reasons To Accept:**

New benchmark and its datasets.

**Reasons To Reject:**

- About effectiveness. For the mentioned four aspects, there are four datasets following the same aspect "semantic", and each other aspect has a dataset. So, is it really enough to evaluate the effectiveness of a newly-proposed model? Especially for your newly-proposed dataset about Stylistic?
- I don't know whether the scale of these datasets is sufficient to form a benchmark. For me, I don't think it's enough, but I will I will draw on the views of other reviewers.

Also,
- Inconsistent statements: Semantic in Table 1 but Content in Table 2
- Not strictly controlling the number of pages

**Reproducibility:**

4: Could mostly reproduce the results, but there may be some variation because of sample variance or minor variations in their interpretation of the protocol or method.

**Reviewer Confidence:**

3: Pretty sure, but there's a chance I missed something. Although I have a good feel for this area in general, I did not carefully check the paper's details, e.g., the math, experimental design, or novelty.

---

> ### Author Rebuttal · Authors · 2023-08-29
>
> We thank the reviewer for their valuable feedback. We will try to address their questions:
>
> Q1: About effectiveness. For the mentioned four aspects, there are four datasets following the same aspect "semantic", and each other aspect has a dataset. So, is it really enough to evaluate the effectiveness of a newly-proposed model? Especially for your newly-proposed dataset about Stylistic?
>
> A1: Our purpose for this benchmark was to propose NLP several setups that provide a natural anomaly detection problem, not an artificially constructed one (e.g. classic text categorization). For example, we propose the following setup for the COLA dataset: we consider a non-grammatically sound sentences as anomalies. In the case of VUA, we consider the metaphoric use of words being the non-regular of employing said words in speech or writing. In the case of Gutenberg Authors, we consider that authorship is a natural problem of anomaly detection: in our setting we consider one author as the main class and we want to distinguish the rest of the texts from his style, said texts being an anomaly from the point of view of his writing style.
>
>
>
> ---
>
> Q2: I don't know whether the scale of these datasets is sufficient to form a benchmark. For me, I don't think it's enough, but I will I will draw on the views of other reviewers.
>
> A2: Regarding the scale of the datasets, it reflects the natural distribution of the data in the wild (online sources). The problems are inherently complex due to fact that there is a scarcity of samples to start from.
>
> ---
>
> Q3: Inconsistent statements: Semantic in Table 1 but Content in Table 2
>
> A3: Thank you, we will fix this in the camera-ready version.
>
> ---
>
> Q4: Not strictly controlling the number of pages
>
> A4: Thank you, we will reorganize the paper such that it strictly follows the size guideline for the camera-ready version.
>
> ---
>
> R5: Can we summarize the relationship between the characteristics of a method and which type of task it is applicable to, based on existing research results?
>
> A5: During our experiments and analysis we observed no correlation between the performance of a model and the type of the task.
>
> We once again thank the reviewer for their insightful feedback and we hope that all of their raised concerns have been addressed.

---

### Official Review · Reviewer_s43E · 2023-08-06

**Soundness:** 2

**Excitement:**

3: Ambivalent: It has merits (e.g., it reports state-of-the-art results, the idea is nice), but there are key weaknesses (e.g., it describes incremental work), and it can significantly benefit from another round of revision. However, I won't object to accepting it if my co-reviewers champion it.

**Paper Topic And Main Contributions:**

The article presents a benchmarking study for anomaly detection in text data. It covers text anomalies at various linguistic levels (syntactic,
semantic, pragmatic, and stylistic) and explores different complexity levels of anomalies. The authors curated three new datasets in addition to the existing corpus. Classical and deep learning methods are evaluated on the benchmark.

**Reasons To Accept:**

The paper is well-written. The topic aligns well with the conference theme and the track, and it is evident that considerable effort has been invested in the research. The article covers text anomalies at various linguistic levels, including syntax, semantics, pragmatics, and style. This inclusive approach provides a comprehensive understanding of anomaly detection in natural language processing.
In addition to widely used corpora like AGNews and 20Newsgroups, the authors have curated three new datasets:
Gutenberg Categories, Song Genres, and Gutenberg Authors. The incorporation of these datasets showcases the authors' commitment
to enhancing the benchmark's diversity and complexity.

**Reasons To Reject:**

The paper presents a few methodological weaknesses that should be addressed.

1. The authors conducted benchmarking on two classical methods, OCSVM and IForest, which are typically considered unsupervised methods. However, as pointed out by Aggarwal 2017, a common mistake in outlier detection benchmarking arises when algorithms depend on user-defined parameters. It is often observed that the algorithms are repeatedly run to select the best parameter, optimizing the ROC AUC score. Unfortunately, such an approach is not acceptable in outlier detection since it effectively incorporates knowledge of the outlier
labels in parameter selection, thereby deviating from the unsupervised nature of the task. In the context of outlier detection, the proper way
to benchmark a pair of algorithms is by evaluating both over a "reasonable" range of parameters and comparing their performance
using a central estimator derived from the resulting runs. Notably, the benchmarking work in anomaly detection usually adheres to this
principle. For instance, Han et al. 2022 employed default hyperparameters in their study.

Aggarwal, C. C. 2017. An introduction to outlier analysis (pp. 134.Springer International Publishing.
Han, S., Hu, X., Huang, H., Jiang, M., & Zhao, Y. 2022. Adbench: Anomaly detection benchmark. Advances in Neural Information Processing Systems, 35, 3214232159.

2) The claim made about grammatically unacceptable phrases (COLA dataset) based on the AUROC score 49.9% is inaccurate . An AUROC score of 50% is equivalent to random guessing, and a score below 50%  indicates that the model's performance is worse than random. Therefore, the claim that grammatically unacceptable phrases represent an anomaly based on this low AUROC score is not
supported. In fact, it suggests that the models are incapable of effectively distinguishing between the classes, and grammatically
unacceptable phrases are not being detected as anomalies.

Moreover, additional information need to be clarified before publication:

3) For classical methods, it is essential to specify the text representation techniques used in anomaly detection. The way text data is represented significantly impacts the anomaly score calculation and, ultimately, the performance of the anomaly detection method. Providing details about the text representation techniques will contribute to a better understanding of
the anomaly detection process for each type of anomaly.

4) The anomaly ratio in the dataset may have a significant impact on the performance of anomaly detection methods. It is crucial
to specify the anomaly ratio for each corpus and each split to provide a comprehensive understanding of the experimental setup and the
results obtained.

**Reproducibility:**

4: Could mostly reproduce the results, but there may be some variation because of sample variance or minor variations in their interpretation of the protocol or method.

**Reviewer Confidence:**

3: Pretty sure, but there's a chance I missed something. Although I have a good feel for this area in general, I did not carefully check the paper's details, e.g., the math, experimental design, or novelty.

---

> ### Author Rebuttal · Authors · 2023-08-29
>
> We thank the reviewer for their valuable feedback. We will try to address their questions:
>
> Q1: The authors conducted benchmarking on two classical methods, OCSVM and IForest, which are typically considered unsupervised methods. However, as pointed out by Aggarwal 2017, a common mistake in outlier detection benchmarking arises when algorithms depend on user-defined parameters. It is often observed that the algorithms are repeatedly run to select the best parameter, optimizing the ROC AUC score. Unfortunately, such an approach is not acceptable in outlier detection since it effectively incorporates knowledge of the outlier labels in parameter selection, thereby deviating from the unsupervised nature of the task. In the context of outlier detection, the proper way to benchmark a pair of algorithms is by evaluating both over a "reasonable" range of parameters and comparing their performance using a central estimator derived from the resulting runs. Notably, the benchmarking work in anomaly detection usually adheres to this principle. For instance, Han et al. 2022 employed default hyperparameters in their study.
>
> A1: While we agree with the reviewer that hyperparameter selection for unsupervised anomaly detection can lead to an overly-optimistic anomaly metric, we would like to argue that it is acceptable for the classical methods (isolation forest, one-class svm) in this case, since our purpose was highlighting the best possible results for them.
>
> Nevertheless, for the neural methods, we don't search for hyperparameters and we use a single set of hyperparameters for all datasets (as mentioned in Sec. 4.1, lines 400-401). Moreover, the neural methods obtain the best results in most of our experiments, therefore we believe that keeping the most optimistic results for the classical baselines is an acceptable approach in this scenatio.
>
> We thank the reviewer for raising this concern, we will update our manuscript with a detailed explanation regarding our hyperparameter selection decisions.
>
> ---
>
> Q2: The claim made about grammatically unacceptable phrases (COLA dataset) based on the AUROC score 49.9% is inaccurate . An AUROC score of 50% is equivalent to random guessing, and a score below 50% indicates that the model's performance is worse than random. Therefore, the claim that grammatically unacceptable phrases represent an anomaly based on this low AUROC score is not supported. In fact, it suggests that the models are incapable of effectively distinguishing between the classes, and grammatically unacceptable phrases are not being detected as anomalies.
>
> A2: While we agree that the results on COLA are just slightly better than random guessing (53.3 AUROC for CVDD), we note that the competing methods have worse than random performance on the dataset, and we believe that having syntactic anomalies for the linguistics domain can be an interesting problem for further work which can employ better inductive biases for this specific scenario.
>
> ---
>
> Q3: For classical methods, it is essential to specify the text representation techniques used in anomaly detection. The way text data is represented significantly impacts the anomaly score calculation and, ultimately, the performance of the anomaly detection method. Providing details about the text representation techniques will contribute to a better understanding of the anomaly detection process for each type of anomaly.
>
> A3: We thank the reviewer for highlighting this issue. For the classical methods, we have used either FastText or GloVe as a feature extraction backbone. Due to space limitations we omitted to report that, but we will update the paper accordingly or we will include this information in the Appendix for the camera-ready version.
>
> ---
>
> Q4: The anomaly ratio in the dataset may have a significant impact on the performance of anomaly detection methods. It is crucial to specify the anomaly ratio for each corpus and each split to provide a comprehensive understanding of the experimental setup and the results obtained.
>
> A4: We have followed the Anomaly Detection setup from [1], [2], in which we pick a class as our inlier dataset, while all of the other classes are treated as outliers. This, indeed, makes the resulting test sets imbalanced. However, we would like to note that our primary metric is AUROC (as opposite to the AUPR metric reported in the Supplementary materials), which is robust to dataset imbalances. Nevertheless, we will report the inlier-outlier ratio for the camera-ready version of the paper.
>
> We once again thank the reviewer for their insightful feedback and we hope that all of their raised concerns have been addressed.

---

### Official Review · Reviewer_vnEh · 2023-08-17

**Soundness:** 3

**Excitement:**

2: Mediocre: This paper makes marginal contributions (vs non-contemporaneous work), so I would rather not see it in the conference.

**Missing References:**

No

**Paper Topic And Main Contributions:**

This paper provides a unified benchmark for anomaly detection in NLP tasks, ranging from syntax to stylistics.  They also build comprehensive datasets, and evaluation metrics and offer several methods for performing anomaly detection.

**Questions For The Authors:**

1. Can you consider more AD approaches, why not only selecting isoforest, oc-svm, cvdd or date?  Can you explain why these methods fit the AD NLP tasks?

2. The evaluation metric, AUROC is used, how about other metrics, e.g., F1 score, can you explain why do you prefer to use AUROC only?

3. Not sure if this work is the first AD benchmark for NLP, can you explain the difference between this work and existing baselines?


**Reasons To Accept:**

The topic is critical to NLP and few studies were provided before. The contribution, specifically.  dataset collection needs much effort.  Broader audience will be interested in this AD benchmark.

**Reasons To Reject:**

The topic is very wide but the context seems to be weak to claim such a topic.  The datasets, either from existing work or newly created collections are still not so strong to cover the major AD tasks and applications.

In addition, the evaluation metrics are not comprehensive to compare the pros/cons of these tasks.  The AD methods are very limited, only four methods are provided. From the results, we observe that most of the best performance is from the DATE method. Not sure how strong about the other baselines, e.g., isoforest, oc-svm, which are very classific AD approaches.

**Reproducibility:**

3: Could reproduce the results with some difficulty. The settings of parameters are underspecified or subjectively determined; the training/evaluation data are not widely available.

**Reviewer Confidence:**

3: Pretty sure, but there's a chance I missed something. Although I have a good feel for this area in general, I did not carefully check the paper's details, e.g., the math, experimental design, or novelty.

**Typos Grammar Style And Presentation Improvements:**

The paper is well-written and easy to follow. No major concern about the writting.

---

> ### Author Rebuttal · Authors · 2023-08-29
>
> We thank the reviewer for their valuable feedback. We will try to address their questions:
>
> Q1: The topic is very wide but the context seems to be weak to claim such a topic. The datasets, either from existing work or newly created collections are still not so strong to cover the major AD tasks and applications.
>
> A1: Our purpose for this benchmark was to propose NLP several setups that provide a natural anomaly detection problem, not an artificially constructed one (e.g. classic text categorization). For example, we propose the following setup for the COLA dataset: we consider a non-grammatically sound sentences as anomalies. In the case of VUA, we consider the metaphoric use of words being the non-regular of employing said words in speech or writing. In the case of Gutenberg Authors, we consider that authorship is a natural problem of anomaly detection: in our setting we consider one author as the main class and we want to distinguish the rest of the texts from his style, said texts being an anomaly from the point of view of his writing style.
>
> ---
>
> Q2: In addition, the evaluation metrics are not comprehensive to compare the pros/cons of these tasks. The AD methods are very limited, only four methods are provided. From the results, we observe that most of the best performance is from the DATE method. Not sure how strong about the other baselines, e.g., isoforest, oc-svm, which are very classific AD approaches.
>
> A2: While we agree with the reviewer that the evaluation metrics were limited, we decided to use AUROC as we consider it the best metric for unbalanced datasets. We also provide extensive AUPR-In and AUPR-Out scores in our Appendix, alongside qualitative figures for CVDD and DATE.
>
> Our methodology is limited due to the fact that anomaly detection in text is not a widely approached subject, thus unexplored. We believe this is an important area of NLP and hope that our benchmark helps the community develop a stronger methodology towards this area or research. The limited analysis is due to the novelty of the problem: we try to categorise possible types of anomalies not apriori (artificially constructed "toy" dataset on which all models would work), but aposteriori, by running the models and reviewing their performance on each class.
>
> ---
>
> Q3: Not sure if this work is the first AD benchmark for NLP, can you explain the difference between this work and existing baselines?
>
> A3: Regarding existing baselines, we've explored the subject in the "Related Work" section of our paper, where we have reported our findings: several frameworks and benchmarks for multitask learning in NLP, but none focused specifically on anomaly detection.
>
> As stated in the answer for the reviewer's first point, our purpose to provide NLP problems that contain a natural anomaly detection setup. Apart from the pre-existing dataset (20NG, AGNews, COLA, VUA), we introduced 3 novel anomaly detection datasets: Gutenberg Authors, Gutengerg Categories and Song Lyrics.
>
> We introduced Gutenberg Authors as an authorship anomaly detection problem, which is still an open problem in NLP. Gutenberg Categories, by comparison, presents more similarities with the other topic detection problems (e.g. 20Newsgroups, AG News), but offers a higher degree of diversity. Same goes for the Song Lyrics dataset. Our purpose with the latter two was to introduce diversity and complexity to the topic anomaly detection field.
>
> We once again thank the reviewer for their insightful feedback and we hope that all of their raised concerns have been addressed.

---

### Official Review · Reviewer_SUCF · 2023-08-19

**Soundness:** 4

**Excitement:**

3: Ambivalent: It has merits (e.g., it reports state-of-the-art results, the idea is nice), but there are key weaknesses (e.g., it describes incremental work), and it can significantly benefit from another round of revision. However, I won't object to accepting it if my co-reviewers champion it.

**Paper Topic And Main Contributions:**

The main contributions of the paper are as follows:
- This paper presents AD-NLP, a benchmark dataset for evaluating anomaly detection methods in natural language processing.
- The benchmark covers 4 anomaly classes - syntactic, semantic, pragmatic, and stylistic. (Table 1)
- The datasets included in the benchmark cover 4 existing datasets that are used in previous literature - 20Newsgroups, AGNews, COLA and VUA.
- The authors further augment this with 3 new proposed datasets based on Project Gutenberg and Song Lyrics data from Textract.
- Then, on the created dataset, 4 methods are evaluated, 2 non-neural (One Class Support Vector Machine, Isolation Forest) and 2 neural (Context Vector Data Description (CVDD), Detecting Anomalies in Text using 452 ELECTRA (DATE)).
- For evaluation, the primary metric considered is AUROC (Area Under the Receiver Operating Characteristic curve). DATE and CVDD usually have the best performance according to this metric across the datasets in the benchmark. (Table 4)

**Reasons To Accept:**

Overall, the goal of having a unified benchmark for anomaly detection in NLP is an important one. The authors highlight some existing challenges and then take an important step towards solving the problem with their proposed benchmark.

Some existing challenges highlighted in the paper and the corresponding solutions proposed:
- Deciding "what is an anomaly" is challenging. (To overcome this they propose 4 anomaly classes - syntactic, semantic, pragmatic and stylistic)
- Very narrow domain of previous datasets and benchmarks (To overcome this, they propose a suite of datasets covering various topics and genres of content).
- The benchmark also overcomes the problem of not having a standard approach for evaluating different methods.

**Reasons To Reject:**

Here are some of my personal opinions about the paper. These reflect things that I would have liked to see but may not necessarily be a strong reason to accept or reject a paper.

- Without available links to anonymized code repositories or a complete description of hyperparameters and other experimental settings, it is not possible for me to verify the reproducibility of the results.
- The motivation for creating a diverse anomaly detection benchmark is reasonable, but could probably be justified a bit better. Why are current evaluation methods insufficient? What specific limitations do they have? For example, explicitly discuss the limitations of current ad-hoc evaluation setups and the benefits of a more comprehensive suite.
- The 3 new datasets created have probably not been introduced as well as they could have been. Reading that section, I saw some details about the construction of the dataset, but very little motivation about why the creation of these particular datasets were necessary and how they are more helpful vs other existing datasets in the public domain. More motivation and less details in this section would have been helpful.
- The analysis and discussion of model performance on different anomaly types could do with some more explorations. In its current state, it is borderline.

**Reproducibility:**

4: Could mostly reproduce the results, but there may be some variation because of sample variance or minor variations in their interpretation of the protocol or method.

**Reviewer Confidence:**

4: Quite sure. I tried to check the important points carefully. It's unlikely, though conceivable, that I missed something that should affect my ratings.

**Typos Grammar Style And Presentation Improvements:**

- This might just be me, but I felt that there was a bit of space wastage in terms of the available 8 pages of the main paper, which could probably have been better utilized to motivate the approach, datasets and analysis more.
- Anonymous GitHub links could have been provided to help with the reproducibility of the results.
- In the first page, after the name of the paper, it says "Anonymous ACL submission" which should be "Anonymous EMNLP submission" but I am not sure if that makes any difference?
- Overall, there are some issues with writing style in the paper that I feel can be resolved with another round of proofreading.

---

> ### Author Rebuttal · Authors · 2023-08-29
>
> We thank the reviewer for their valuable feedback. We will try to address their questions:
>
> R1: Without available links to anonymized code repositories or a complete description of hyperparameters and other experimental settings, it is not possible for me to verify the reproducibility of the results.
>
> A1: Thank you for pointing this out, we will add the anonymized code to the camera-ready version of the paper, but we also provide it here: https://anonymous.4open.science/r/ad-nlp-B01C/README.md.
>
> ---
>
> R2: The motivation for creating a diverse anomaly detection benchmark is reasonable, but could probably be justified a bit better. Why are current evaluation methods insufficient? What specific limitations do they have? For example, explicitly discuss the limitations of current ad-hoc evaluation setups and the benefits of a more comprehensive suite.
>
> A2: Our purpose for this benchmark was to propose NLP several setups that provide a natural anomaly detection problem, not an artificially constructed one (e.g. classic text categorization). For example, we propose the following setup for the COLA dataset: we consider a non-grammatically sound sentences as anomalies. In the case of VUA, we consider the metaphoric use of words being the non-regular of employing said words in speech or writing. In the case of Gutenberg Authors, we consider that authorship is a natural problem of anomaly detection: in our setting we consider one author as the main class and we want to distinguish the rest of the texts from his style, said texts being an anomaly from the point of view of his writing style.
>
> ---
>
> R3: The 3 new datasets created have probably not been introduced as well as they could have been. Reading that section, I saw some details about the construction of the dataset, but very little motivation about why the creation of these particular datasets were necessary and how they are more helpful vs other existing datasets in the public domain. More motivation and less details in this section would have been helpful.
>
> A3: We introduced Gutenberg Authors as an authorship anomaly detection problem, which is still an open problem in NLP. Gutenberg Categories, by comparison, presents more similarities with the other topic detection problems (e.g. 20Newsgroups, AG News), but offers a higher degree of diversity. Same goes for the Song Lyrics dataset. Our purpose with the latter two was to introduce diversity and complexity to the topic anomaly detection field.
>
> ---
>
> R4: The analysis and discussion of model performance on different anomaly types could do with some more explorations. In its current state, it is borderline.
>
>
> A4: We offer more detailed quantitative results (AUPR-In , AUPR-Out scores) in the Appendix, as well as a couple of qualitative analysis plots of the deep learning approaches. We will add more comments on the results we provided as part of the Appendix.
>
> ---
>
> R5: In the first page, after the name of the paper, it says "Anonymous ACL submission" which should be "Anonymous EMNLP submission" but I am not sure if that makes any difference?
>
> A5: Thank you for pointing this out! We started writing the paper with an ACL template format. At the end we changed to the EMNLP format and forgot to change the subtitle of the paper to “Anonymous EMNLP submission”. We will do so for the camera-ready version.
>
> ---
>
> R6: Overall, there are some issues with writing style in the paper that I feel can be resolved with another round of proofreading.
>
> A6: Thank you! We will do another round of proof-reading for the camera-ready version.
>
> We once again thank the reviewer for their insightful feedback and we hope that all of their raised concerns have been addressed.

---

### Meta-Review · Area_Chair_Fx5G · 2023-09-10

**Recommendation:** 5

**Metareview:**

This paper presents a new benchmark for evaluating anomaly detection methods in natural language processing, called AD-NLP. This benchmark includes 4 existing datasets and adds 3 new datasets. All the data are organized following a classification of anomality (syntactic, semantic, pragmatic, and stylistic). Moreover different methods are evaluated on this benchmark. The benchmark will be made available.

It would be great to better present the results in the paper instead of in appendix. Also more explanation and information are need at some point (choice of hyperparameters for the NN methods and representations methods).

---

### Decision · Program_Chairs · 2023-10-07

**Decision:**

Accept-Main

**Comment:**

This paper presents a new benchmark for evaluating anomaly detection methods in natural language processing, called AD-NLP. This benchmark includes 4 existing datasets and adds 3 new datasets. All the data are organized following a classification of anomality (syntactic, semantic, pragmatic, and stylistic). Moreover different methods are evaluated on this benchmark. The benchmark will be made available.

It would be great to better present the results in the paper instead of in appendix. Also more explanation and information are need at some point (choice of hyperparameters for the NN methods and representations methods).